# Accuracy of rapid point-of-care antigen-based diagnostics for SARS-CoV-2: An updated systematic review and meta-analysis with meta-regression analyzing influencing factors

Lukas E. Brümmer[1], Stephan Katzenschlager[2], Sean McGrath[3], Stephani Schmitz[4], Mary Gaeddert[1], Christian Erdmann[5], Marc Bota[6], Maurizio Grilli[7], Jan Larmann[2], Markus A. Weigand[2], Nira R. Pollock[8], Aurélien Macé[9], Berra Erkosar[9], Sergio Carmona[9], Jilian A. Sacks[9], Stefano Ongarello[9], Claudia M. Denkinger[1,10]*

**1** Division of Infectious Disease and Tropical Medicine, Center for Infectious Diseases, Heidelberg University Hospital, Heidelberg, Germany, **2** Department of Anesthesiology, Heidelberg University Hospital, Heidelberg, Germany, **3** Department of Biostatistics, Harvard T.H. Chan School of Public Health, Boston, Massachusetts, United States of America, **4** Department of Developmental Biology, Erasmus Medical Center, Rotterdam, the Netherlands, **5** FH Muenster University of Applied Sciences, Muenster, Germany, **6** Agaplesion Bethesda Hospital, Hamburg, Germany, **7** Library, University Medical Center Mannheim, Mannheim, Germany, **8** Department of Laboratory Medicine, Boston Children's Hospital, Boston, Massachusetts, United States of America, **9** FIND, Geneva, Switzerland, **10** German Center for Infection Research (DZIF), partner site Heidelberg University Hospital, Heidelberg, Germany

☯ These authors contributed equally to this work.

* Claudia.Denkinger@uni-heidelberg.de

**Data Availability Statement:** All data are available from https://doi.org/10.11588/data/T3MIB0.

## Abstract

### Background

Comprehensive information about the accuracy of antigen rapid diagnostic tests (Ag-RDTs) for Severe Acute Respiratory Syndrome Coronavirus 2 (SARS-CoV-2) is essential to guide public health decision makers in choosing the best tests and testing policies. In August 2021, we published a systematic review and meta-analysis about the accuracy of Ag-RDTs. We now update this work and analyze the factors influencing test sensitivity in further detail.

### Methods and findings

We registered the review on PROSPERO (registration number: CRD42020225140). We systematically searched preprint and peer-reviewed databases for publications evaluating the accuracy of Ag-RDTs for SARS-CoV-2 until August 31, 2021. Descriptive analyses of all studies were performed, and when more than 4 studies were available, a random-effects meta-analysis was used to estimate pooled sensitivity and specificity with reverse transcription polymerase chain reaction (RT-PCR) testing as a reference. To evaluate factors influencing test sensitivity, we performed 3 different analyses using multivariable mixed-effects meta-regression models. We included 194 studies with 221,878 Ag-RDTs performed. Overall, the pooled estimates of Ag-RDT sensitivity and specificity were 72.0% (95% confidence interval [CI] 69.8 to 74.2) and 98.9% (95% CI 98.6 to 99.1). When manufacturer instructions

**Funding:** The study was supported by the Ministry of Science, Research and Arts of the State of Baden-Wuerttemberg, Germany (no grant number; https://mwk.badenwuerttemberg.de/de/startseite/) and internal funds from the Heidelberg University Hospital (no grant number; https://www.heidelberg-university-hospital.com/de/) to CMD. Further, this project was funded by United Kingdom (UK) aid from the British people (grant number: 300341-102; Foreign, Commonwealth & Development Office (FCMO), former UK Department of International Development (DFID); www.gov.uk/fcdo), and supported by a grant from the World Health Organization (WHO; no grant number; https://www.who.int) and a grant from Unitaid (grant number: 2019-32-FIND MDR; https://unitaid.org) to Foundation of New Diagnostics (FIND; JAS, SC, SO, AM, BE). This study was also funded by the National Science Foundation GRFP (grant number DGE1745303) to SM. For the publication fee we acknowledge financial support by Deutsche Forschungsgemeinschaft within the funding programme „Open Access Publikationskosten" (no grant number; https://www.dfg.de/en/index.jsp), as well as by Heidelberg University (no grant number; https://www.uni-heidelberg.de/en). The funders had no role in study design, data collection and analysis, decision to publish, or preparation of the manuscript.

**Competing interests:** I have read the journal's policy and the authors of this manuscript have the following competing interests: CMD is a member of the Editorial Board of PLOS Medicine. CMD also declares a payment from Roche Diagnostics that she accepted as German law requires a manufacturer to pay for the use of data for regulatory purposes. Data was generated as part of an independent evaluation by CMD and team. AM, BE, SC, JAS and SO are employees of FIND, the global alliance for diagnostics.

**Abbreviations:** Ag-RDT, antigen rapid diagnostic test; AN, anterior nasal; CI, confidence interval; COVID-19, Coronavirus Disease 2019; Ct, cycle threshold; FN, false negative; IFU, instructions for use; IQR, interquartile range; MT, mid-turbinate; NP, nasopharyngeal; OP, oropharyngeal; POC, point-of-care; RT-PCR, reverse transcription polymerase chain reaction; SARS-CoV-2, Severe Acute Respiratory Syndrome Coronavirus 2; sROC, Summary receiver operating characteristic; TP, true positive; VoC, variants of concern.

were followed, sensitivity increased to 76.3% (95% CI 73.7 to 78.7). Sensitivity was markedly better on samples with lower RT-PCR cycle threshold (Ct) values (97.9% [95% CI 96.9 to 98.9] and 90.6% [95% CI 88.3 to 93.0] for Ct-values <20 and <25, compared to 54.4% [95% CI 47.3 to 61.5] and 18.7% [95% CI 13.9 to 23.4] for Ct-values $\geq$25 and $\geq$30) and was estimated to increase by 2.9 percentage points (95% CI 1.7 to 4.0) for every unit decrease in mean Ct-value when adjusting for testing procedure and patients' symptom status. Concordantly, we found the mean Ct-value to be lower for true positive (22.2 [95% CI 21.5 to 22.8]) compared to false negative (30.4 [95% CI 29.7 to 31.1]) results. Testing in the first week from symptom onset resulted in substantially higher sensitivity (81.9% [95% CI 77.7 to 85.5]) compared to testing after 1 week (51.8%, 95% CI 41.5 to 61.9). Similarly, sensitivity was higher in symptomatic (76.2% [95% CI 73.3 to 78.9]) compared to asymptomatic (56.8% [95% CI 50.9 to 62.4]) persons. However, both effects were mainly driven by the Ct-value of the sample. With regards to sample type, highest sensitivity was found for nasopharyngeal (NP) and combined NP/oropharyngeal samples (70.8% [95% CI 68.3 to 73.2]), as well as in anterior nasal/mid-turbinate samples (77.3% [95% CI 73.0 to 81.0]). Our analysis was limited by the included studies' heterogeneity in viral load assessment and sample origination.

## Conclusions

Ag-RDTs detect most of the individuals infected with SARS-CoV-2, and almost all (>90%) when high viral loads are present. With viral load, as estimated by Ct-value, being the most influential factor on their sensitivity, they are especially useful to detect persons with high viral load who are most likely to transmit the virus. To further quantify the effects of other factors influencing test sensitivity, standardization of clinical accuracy studies and access to patient level Ct-values and duration of symptoms are needed.

## Author summary

### Why was this study done?

- Antigen rapid diagnostic tests (Ag-RDTs) for Severe Acute Respiratory Syndrome Coronavirus 2 (SARS-CoV-2) have proven to be a cornerstone in the fight against the Coronavirus Disease 2019 (COVID-19) pandemic.

- In an earlier analysis, we found Ag-RDTs to be 76.3% sensitive and 99.1% specific, but with sensitivity varying between test manufacturers, the way tests were performed, and the patients in which they were used.

- We now present an updated analysis and explore the factors influencing Ag-RDTs' sensitivity and driving heterogeneity in the results in further detail.

## What did the researchers do and find?

- We searched multiple preprint and peer-reviewed databases for clinical accuracy studies evaluating Ag-RDTs for SARS-CoV-2 at the point of care.

- Ag-RDTs proved to be 76.3% (95% confidence interval (CI) 73.7 to 78.7) sensitive and 99.1% (95% CI 98.8 to 99.3) specific, when performed as per the manufacturer's instructions.

- Sensitivity increased by 2.9 percentage points (95% CI 1.7 to 4.0) for each unit the mean cycle threshold (Ct)-value, a semiquantitative measurement of the real-time polymerase chain reaction test decreased, and sensitivity was highest in samples with a Ct-value <20 (i.e., a high viral load; sensitivity of 97.9% [95% CI 96.9 to 98.9]).

- Higher sensitivity was also found in samples originating from symptomatic compared to asymptomatic persons, especially when study participants were still within the first week of symptom onset, but these effects were mainly driven by the sample's viral load.

## What do these findings mean?

- Compared to our previous analysis, Ag-RDTs continue to show high sensitivity and excellent specificity in detecting SARS-CoV-2.

- With viral load being the main driver behind test sensitivity, Ag-RDTs detect almost all of the persons with high viral load, who are at the greatest risk of transmitting the virus.

- While it is unlikely that the overall performance of Ag-RDTs will substantially change, further research is needed to analyze the accuracy of Ag-RDTs for different virus variants and sample types, as well as methods of test performance (e.g., self-performed, instrument based) in more detail.

## Introduction

Antigen rapid diagnostic tests (Ag-RDTs) for Severe Acute Respiratory Syndrome Coronavirus 2 (SARS-CoV-2) have proven to be a cornerstone in fighting the Coronavirus Disease 2019 (COVID-19) pandemic, as they provide results quickly and are easy to use [1]. Nevertheless, the Ag-RDTs' performance differs widely between manufacturers, the way they are performed, and the patients in which they are used [2,3]. Thus, a comprehensive synthesis of evidence on commercially available Ag-RDTs and the factors influencing their accuracy is vital to guide public health decision makers in choosing the right test for their needs [4].

Starting in October 2020, we conducted a living systematic review (available online at www.diagnosticsglobalhealth.org, updated weekly until August 31, 2021), summarizing the accuracy of commercially available Ag-RDTs reported in scientific literature. To equip public health decision makers with the latest findings, we published the results of our first review as soon as possible in February 2021 (including literature until December 15, 2020) [5]. After peer-review and including studies for 4 further months (until April 30, 2021), we published an updated review. Here, when performed as per the manufacturer's instructions, pooled estimates of Ag-

RDT sensitivity and specificity were 76.3% (95% confidence interval (CI) 73.1% to 79.2%) and 99.1% (95% CI 98.8% to 99.4). The most sensitive test was the SARS-CoV-2 Antigen Test (LumiraDx, United Kingdom; henceforth called LumiraDx) [4].

Since our last update, many additional studies have been published with a substantial increase in studies assessing asymptomatic participants, allowing for further sub-analysis of findings [3,6]. In addition, we and others found Ag-RDT sensitivity to decrease significantly in persons with lower viral load. Viral load is usually estimated through the number of cycles, i.e., the cycle threshold (Ct) value, a reverse transcriptase polymerase chain reaction (RT-PCR) needs to be performed until viral RNA can be detected, with a low Ct-value indicating a high viral load. Furthermore, sensitivity decreased in asymptomatic persons or persons with more than 7 days since symptom onset (DOS > 7) [4]. However, studies including symptomatic patients enroll persons typically within days since onset of symptoms [7], when viral load is highest [8,9]. On the contrary, studies including only asymptomatic persons have a higher chance of including persons at a later stage in the disease and thus with lower viral load. Therefore, the decrease in Ag-RDT sensitivity might only be driven by viral load, irrespective of persons' symptom status.

With the present work, we aim not only to give an updated overview on the accuracy of commercially available Ag-RDTs, but also to further explore the impact of viral load, the presence of symptoms, and testing procedure on the accuracy of Ag-RDTs.

## Methods

We developed a study protocol following standard guidelines for systematic reviews [10,11], which is available in the Supporting information (S1 Text). We also completed the PRISMA checklist (S1 PRISMA Checklist) and registered the review on PROSPERO (registration number: CRD42020225140).

### Search strategy

We performed a search of the databases PubMed, Web of Science, medRxiv, and bioRxiv. The search terms were developed with an experienced medical librarian (MG), using combinations of subject headings (when applicable) and text-words for the concepts of the search question, and checked against an expert assembled list of relevant papers. The main search terms were "Severe Acute Respiratory Syndrome Corona-virus 2," "COVID-19," "Betacoronavirus," "Coronavirus," and "Point of Care Testing" with no language restrictions. The full list of search terms is available in S2 Text. Also, 1 author (LEB) manually searched the website of FIND, the global alliance for diagnostics (https://www.finddx.org/sarscov2-eval-antigen/), for additional relevant studies, and search results were checked by a second author (SK). We performed the search biweekly through August 31, 2021. The last manual search of the FIND website was performed on September 10, 2021. In addition to conducting the present review, we updated our website www.diagnosticsglobalhealth. org weekly with the latest search results based on the methods outlined below.

### Inclusion criteria

We included studies evaluating the accuracy of commercially available Ag-RDTs to establish a diagnosis of SARS-CoV-2 infection at the point-of-care (POC), against RT-PCR or cell culture as reference standard. We included all study populations irrespective of age, presence of symptoms, or study location. No language restrictions were applied. We considered cohort studies, nested cohort studies, case–control or cross-sectional studies, and randomized studies. We included both peer-reviewed publications and preprints.

We excluded studies, in which patients were tested for the purpose of monitoring or ending quarantine. Also, publications with a population size smaller than 10 were excluded (although

the size threshold of 10 is arbitrary, such small studies are more likely to give unreliable estimates of sensitivity and specificity). Analytical accuracy studies, where tests are performed on spiked samples with a known quantity of virus, were also excluded.

### Index tests

Ag-RDTs for SARS-CoV-2 aim to detect infection by recognizing viral proteins (typically the SARS-CoV-2 nucleoprotein). Most Ag-RDTs dedicated for POC deployment use specific labeled antibodies attached to a nitrocellulose matrix strip (lateral flow assay) to capture and detect the viral antigen. Successful binding of the antigen to the antibodies is either detected visually by the appearance of a line on the matrix strip or through a specific reader instrument for fluorescence detection. Other POC instrument-based tests use chips or cartridges that enable an automated immunoassay testing procedure. Ag-RDTs typically provide results within 10 to 30 minutes [3].

### Reference standard

Viral culture detects viable virus that is relevant for transmission but is only available in research settings. Since RT-PCR tests are more widely available and SARS-CoV-2 RNA (as reflected by RT-PCR Ct-value) highly correlates with SARS-CoV-2 antigen quantities [12], we considered RT-PCR an acceptable reference standard for the purposes of this systematic review. Where an international standard for the correlation of the viral load to the Ct-values was used, we also report the viral load [13].

### Study selection and data extraction

Two reviewers (LEB and SS, LEB and CE, or LEB and MB) reviewed the titles and abstracts of all publications identified by the search algorithm independently, followed by a full-text review of those eligible, to select the articles for inclusion in the systematic review. Any disputes were solved by discussion or by a third reviewer (CMD).

Studies that assessed multiple Ag-RDTs or presented results based on differing parameters (e.g., various sample types) were considered as individual data sets. At first, 4 authors (SK, CE, SS, and MB) extracted 5 randomly selected papers in parallel to align data extraction methods. Afterwards, data extraction and the assessment of methodological quality and independence from test manufacturers (see below) were performed by 1 author per paper (LEB, SK, CE, SS, or MB) and reviewed by a second (LEB, SK, SS, or MB). Any differences were resolved by discussion or by consulting a third author (CMD). The data items extracted can be found in the Supporting information (S1 Table).

### Assessment of methodological quality

The quality of the clinical accuracy studies was assessed by applying the QUADAS-2 tool [14]. The tool evaluates 4 domains: study participant selection, index test, reference standard, and flow and timing. For each domain, the risk of bias is analyzed using different signaling questions. Beyond the risk of bias, the tool also evaluates the applicability of each included study to the research question for every domain. We prepared a QUADAS-2 assessment guide specific to the needs of this review, which can be found in the Supporting information (S3 Text).

### Assessment of independence from manufacturers

We examined whether a study received financial support from a test manufacturer (including the free provision of Ag-RDTs), whether any study author was affiliated with a test

manufacturer, and whether a respective conflict of interest was declared. Studies were judged not to be independent from the test manufacturer if at least 1 of these aspects was present; otherwise, they were considered to be independent.

## Statistical analysis and data synthesis

We extracted raw data from the studies and recalculated performance estimates where possible based on the extracted data. Also, some primary studies reported the median Ct-value along with the first and third interquartile range (IQR) and/or minimum and maximum values rather than the sample mean and standard deviation. To incorporate these studies in our analyses, we applied the quantile estimation approach [15] to estimate the mean and standard deviation of the Ct-values. In an effort to use as much of the heterogeneous data as possible, the cutoffs for the Ct-value groups were relaxed by 2 to 3 points within each range. The <20 group included values reported up to ≤20, the <25 group included values reported as ≤24 or <25 or 20 to 25, and the <30 group included values from ≤29 to ≤33 and 25 to 30. The ≥25 group included values reported as ≥25 or 25 to 30, and the ≥30 group included values from ≥30 to ≥35. For the same reason, when categorizing by age, the age group <18 years (children) included samples from persons whose age was reported as <16 or <18 years, whereas the age group ≥18 years (adults) included samples from persons whose age was reported as ≥16 or ≥18 years. Also, for the symptom duration groups, the ≤7 days group included ≤4, ≤5, ≤6, 6 to 7, ≤7, and ≤9 days, and the >7 days group included >5, 6 to 10, 6 to 21, >7, and 8 to 14 days. Relaxing the boundaries for the Ct-value, age, and duration of symptoms subgroup resulted in some overlap within the respective groups. Predominant variants of concern (VoC) for each study were analyzed using the online tool CoVariants [16] with respect to the stated study period. The respective VoCs were extracted according the current WHO listing [17]. The raw data can be found in the Supporting information (S2 Table) and with more details online (https://doi.org/10.11588/data/T3MIB0).

If 4 or more data sets were available with at least 20 RT-PCR-positive samples per data set for a predefined analysis, a meta-analysis was performed. We report pooled estimates of sensitivity and specificity for SARS-CoV-2 detection along with 95% CIs using a bivariate model (implemented with the "reitsma" command from the R package "mada," version 0.5.10). Summary receiver operating characteristic (sROC) curves were created for the 2 Ag-RDTs with the highest sensitivity. In subgroup analyses (below), where papers presented data only on sensitivity, a univariate random effects inverse variance meta-analysis was performed (using the "metagen" command from the R package "meta," version 5.1–1, and the "rma" command from the R package "metafor," version 3.0–2). When there were fewer than 4 studies for an index test, only a descriptive analysis was performed, and accuracy ranges are reported.

We prepared forest plots for the sensitivity and specificity of each test and visually evaluated the heterogeneity between studies. In addition, heterogeneity was assessed by calculating Cochran's Q and I2 indices. Because there is no standard method taking into account the correlation between the sensitivity and specificity in bivariate models, we calculated these indices from a pooled diagnostic odds ratio using the "madauni" function from the "mada" package. However, while this was the only approach possible, we do not view it as fully statistical stringent and present the resulting Cochran's Q and $I^2$ only in the Supporting information (S3 Table). For the univariate models, the heterogeneity measures were obtained from the "metagen" model output directly and are reported in the results section.

We predefined subgroups for meta-analysis based on the following characteristics: Ct-value range, testing procedure in accordance with manufacturer's instructions as detailed in the instructions for use (IFU) (henceforth called IFU-conforming) versus not IFU-conforming,

age (<18 versus ≥18 years), sample type, presence or absence of symptoms, symptom duration (≤7 days versus >7 days), viral load, and predominant SARS-CoV-2 variant. We also provide mean Ct-value across true positive (TP) and false negative (FN) test results. For categorization by sample type, we assessed (1) nasopharyngeal (NP) alone or combined with other (e.g., oro-pharyngeal [OP]); (2) OP alone; (3) anterior nasal (AN) or mid-turbinate (MT); (4) a combination of bronchoalveolar lavage and throat wash (BAL/TW); or (5) saliva.

We applied multivariable linear mixed-effect meta-regression models to explore factors that affect diagnostic test sensitivity. Based on our previous analysis [4], we a priori defined an individual's time since infection and sample type and condition as underlying factors, influencing test sensitivity through an individual's symptom status (symptomatic versus asymptomatic), the sample's viral load (estimated by the mean Ct-value as presented in the study for the sub cohort of interest), and the testing procedure (IFU- versus not IFU-conforming). We performed 3 different analyses, each of which obtained unadjusted and adjusted estimates (i.e., an estimate of the association between a factor and test sensitivity, holding the other covariates in the model constant) of the effect of factors on test sensitivity.

In the first analysis, we estimated the direct effect of symptom status, viral load, and testing procedure on test sensitivity. For the second and third analysis, we restricted the meta-regression models to data sets of symptomatic persons due to a lack of data. Specifically, the second analysis assessed the effect of time since infection (estimated as the sample mean of symptom duration), viral load, and testing procedure on test sensitivity. The third analysis also assessed the effect of time since infection, viral load, and testing procedure on test sensitivity, but depicted the time since infection as a binary covariate of the symptom duration subgroup (≤7 versus >7 days). Further details on the implementation of the meta-regression models and the underlying casual diagrams are available in the Supporting information (Figs A and B in S4 Text). Data sets with less than 5 RT-PCR positives were excluded. We considered an effect to be statistically significant when the regression coefficient's 95% CI did not include 0. The analyses were performed using the "metafor" R package, version 3.0–2 [18].

As recommended to investigate publication bias for diagnostic test accuracy meta-analyses, we performed the Deeks' test for funnel-plot asymmetry [19] (using the "midas" command in Stata, version 15); a $p$-value < 0.10 for the slope coefficient indicates significant asymmetry.

## Sensitivity analysis

Three sensitivity analyses were performed: estimation of sensitivity and specificity excluding case–control studies, estimation of sensitivity and specificity excluding not peer-reviewed studies, and estimation of sensitivity and specificity excluding studies that were potentially influenced through test manufacturers. We compared the results of each sensitivity analysis against the overall results to assess the potential bias introduced by case–control, not peer-reviewed, and manufacturer-influenced studies.

## Results

### Summary of studies

The systematic search resulted in 31,254 articles. After removing duplicates, 11,462 articles were screened, and 433 papers were considered eligible for full-text review. Of these, 259 were excluded because they did not present primary data or the Ag-RDT was not commercially available. For similar reasons, we also excluded 4 studies from the FIND website. A list of the studies excluded and their reason for exclusion can be found in the Supporting information (S5 Text). This left 174 studies from the systematic search [20–193] as well as further 20 studies from the FIND website [194–213] to be included in the review (Fig 1).

At the end of the data extraction process, 21 studies were still in preprint form [20,21,25,51,54,59,62,69,73,78,88,104,120,125,133,164,171,172,177,178,190]. All studies included were written in English, except for 3 in Spanish [57,66,138], 1 in Turkish [99], and 1 in French [157]. Out of the 194 studies, 26 conducted a case–control study [25,36,38,70,71,76,85,88,92–94,97,98,100,107,112,139,144,148,149,155,160,170,172,186,188], while the remaining 168 were cross-sectional or cohort studies. The reference method was RT-PCR in all except 1 study, which used viral culture [139].

The 194 studies were divided into 333 data sets. Across these, 76 different Ag-RDTs were evaluated (75 lateral flow assays, of which 63 are interpreted visually and 12 required an auto-mated, proprietary reader; 1 assay is an automated immunoassay). The most common reasons for testing were the occurrence of symptoms (98 data sets, 29.4% of data sets) and screening of asymptomatic persons with (3; 0.9%) or without (22; 6.6%) close contact to a SARS-CoV-2 confirmed case. In 142 (42.6%) of the data sets, individuals were tested due to more than 1 of the reasons mentioned and for 68 (20.4%) the reason for testing was unclear.

In total, 221,878 Ag-RDTs were performed, with a mean number of samples per data set of 666 (range 15 to 22,994). The age of the individuals tested was specified for only 90,981 samples, of which 84,119 (92.5%) were from adults (age group ≥18) and 6,862 (7.5%) from children (age group <18). Symptomatic persons comprised 74,118 (33.4%) samples, while 97,982 (44.2%) samples originated from asymptomatic persons, and for 49,778 (22.4%) samples, the participant's symptom status was not stated by the authors. The most common sample type evaluated was NP and mixed NP/OP (117,187 samples, 52.8%), followed by AN/MT (86,354 samples, 38.9%). There was substantially less testing done on the other sample types, with 3,586 (1.6%) tests done from OP samples, 1,256 (0.6%) from saliva, 219 (0.1%) from BAL/TW, and for 13,276 (6.0%) tests the type of sample was not specified in the respective studies.

A summary of the tests evaluated in clinical accuracy studies, including study author and sample size, as well as study design aspects that could potentially influence test performance, such as sample type, sample condition, IFU conformity, and symptom status, can be found in the Supporting information (S2 Table). The Standard Q test (SD Biosensor, South Korea; dis-tributed in Europe by Roche, Germany; henceforth called Standard Q) was the most frequently used with 57 (17.1%) data sets and 36,246 (16.3%) tests, while the Panbio test (Abbott Rapid Diagnostics, Germany; henceforth called Panbio) was assessed in 55 (16.5%) data sets with 38,620 (17.4%) tests performed. Detailed results for each clinical accuracy study are available in the Supporting information (S1 Fig).

## Methodological quality of studies

The findings on study quality using the QUADAS-2 tool are presented in Fig 2A and 2B. In 294 (88.3%) data sets, a relevant study population was assessed. However, for only 68 (20.4%) of the data sets, the selection of study participants was considered representative of the setting and population chosen (i.e., they avoided inappropriate exclusions or a case–control design and enrollment occurred consecutive or randomly).

The conduct and interpretation of the index tests were considered to have low risk of bias in 176 (52.9%) data sets (e.g., through appropriate blinding of persons interpreting the visual read-out). However, for 155 (46.5%) data sets, sufficient information to clearly judge the risk of bias was not provided. In only 151 (45.3%) data sets, the Ag-RDTs were performed accord-ing to IFU, while 138 (41.4%) were not IFU-conforming, potentially impacting the diagnostic accuracy; for 44 (13.2%) data sets, the IFU status was unclear. The most common deviations from the IFU were (1) use of samples that were prediluted in transport media not recom-mended by the manufacturer (113 data sets, 12 unclear); (2) use of banked samples (103 data

**PRISMA 2020 flow diagram for new systematic reviews which included searches of databases, registers and other sources**

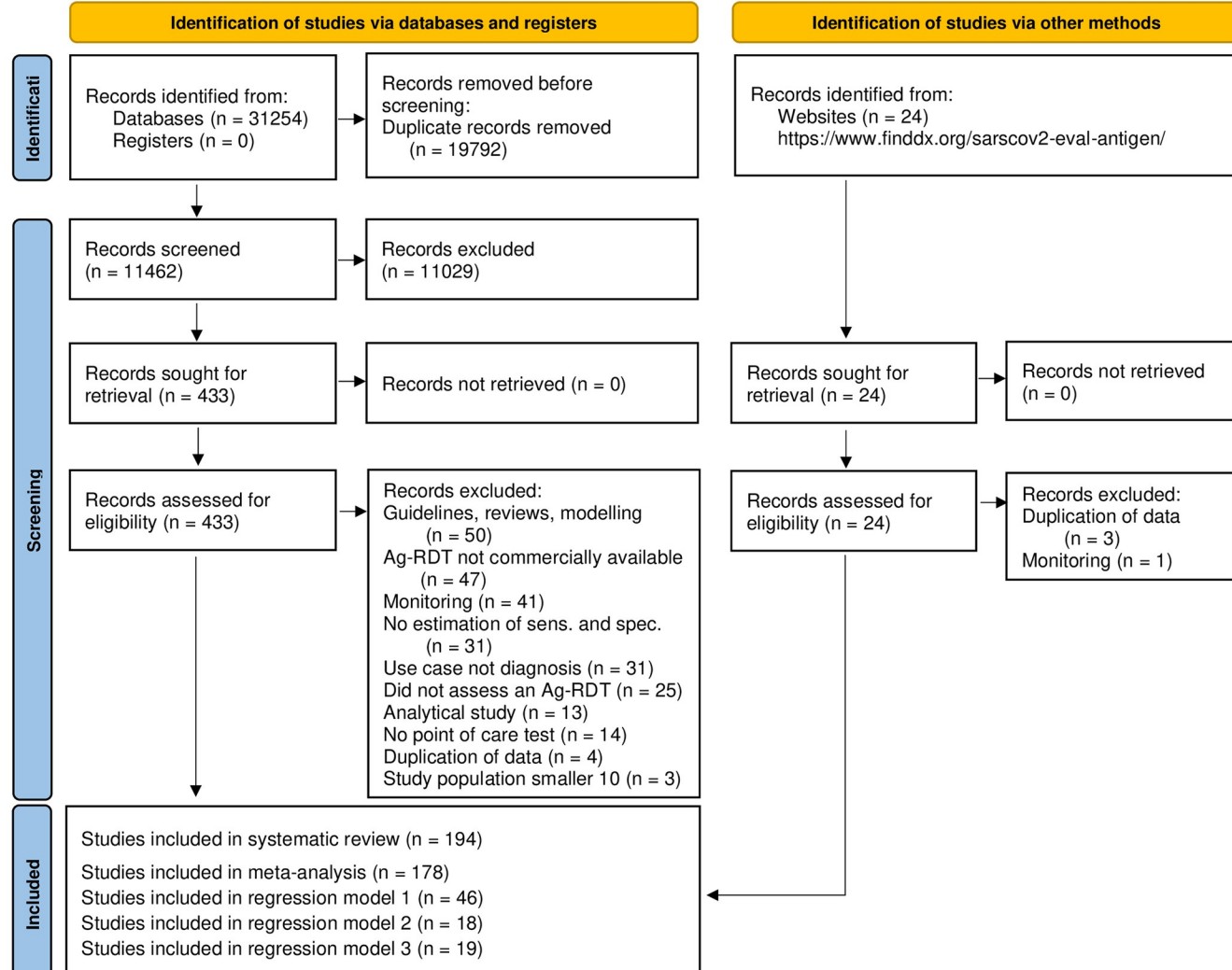

**Fig 1. PRISMA flow diagram.** Based on Page and colleagues [214]. Ag-RDT, antigen rapid diagnostic tests; IFU, instructions for use.

sets, 12 unclear); and (3) a sample type that was not recommended for Ag-RDTs (8 data sets, 11 unclear).

In 126 (37.8%) data sets, the reference standard was performed before the Ag-RDT, or the operator conducting the reference standard was blinded to the Ag-RDT results, resulting in a low risk of bias. In almost all other data sets (206; 61.9%), this risk could not be assessed, due to missing information and for 1 data set (0.3%) intermediate concern was raised. The applicability of the reference test was judged to be of low concern for all data sets, as viral culture or RT-PCR are considered to adequately define the target condition for the purpose of this study.

In 327 (98.2%) data sets, the sample for the index test and reference test were obtained at the same time, while this was unclear in 6 (1.8%). In 227 (68.2%) data sets, the same RT-PCR assay was used as the reference of all included samples, while in 85 (25.5%) data sets, multiple RT-PCR assays were used as the reference. The RT-PCR systems used most frequently were the Cobas SARS-CoV-2 Test (Roche, Germany; used in 79 data sets [23.7%]), the Allplex 2019-nCoV Assay (Seegene, South Korea; used in 61 data sets [18.3%]), and the GeneXpert

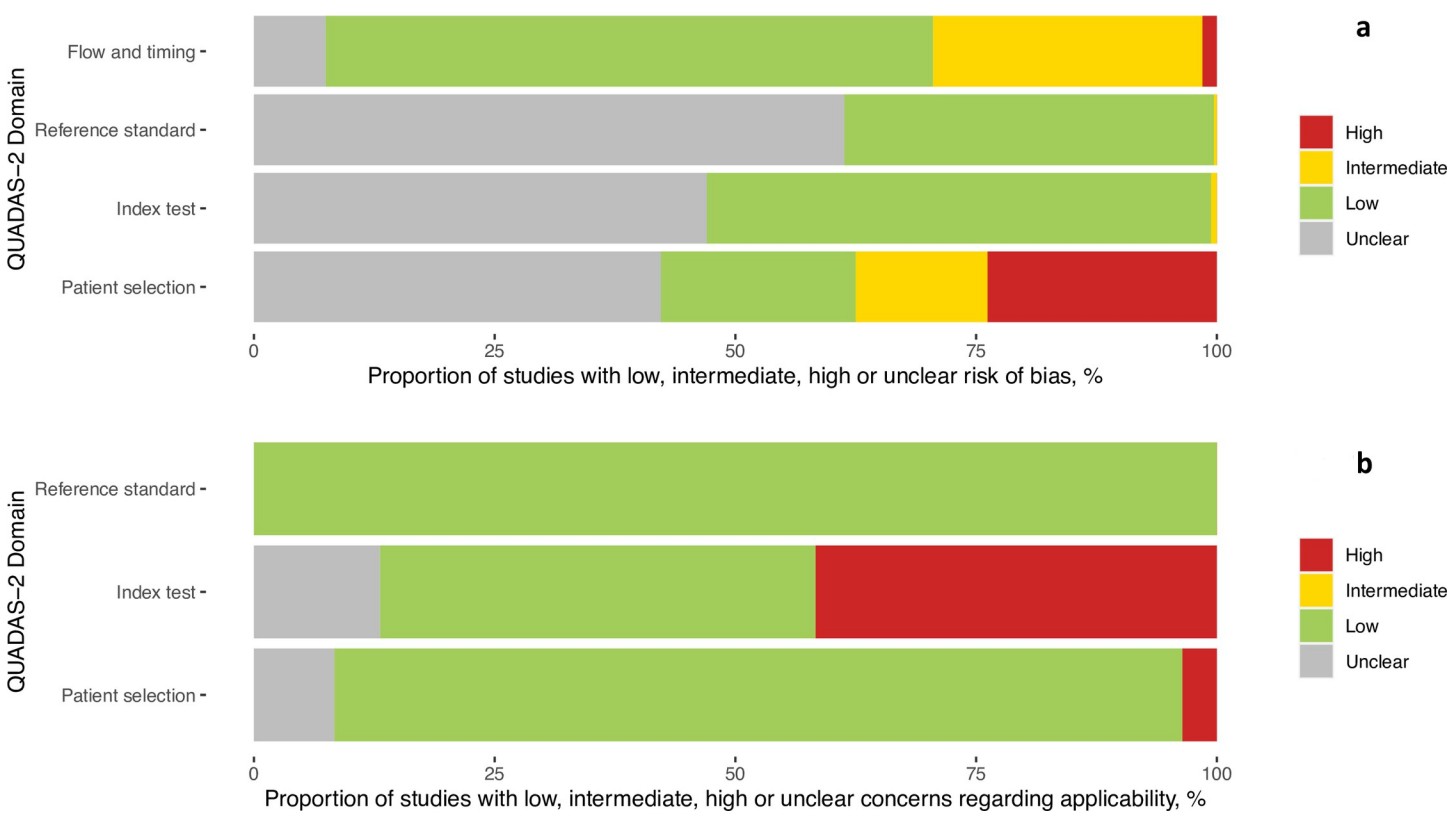

**Fig 2.** (a) Methodological quality of the clinical accuracy studies (risk of bias). (b) Methodological quality of the clinical accuracy studies (applicability).

(Cepheid, United States, CA; used in 34 data sets [10.2%]). For 21 (6.3%) data sets, the RT-PCR used as reference standard was unclear. The RT-PCR system, its limit of detection (if publicly available from the manufacturer) and sample type used in each data set can be found in the Supporting information (S2 Table). Furthermore, for 19 (5.7%) data sets, there was a concern that not all selected study participants were included in the analysis.

Finally, 45 (23.2%) of the studies received financial support from the Ag-RDT manufacturer. In 13 of these as well as in 2 others (in total 7.7% of all studies), employment of the authors by the manufacturer of the Ag-RDT studied was indicated. The respective studies are listed in the Supporting information (S6 Text). Overall, a competing interest was found in 47 (24.2%) of the studies. Detailed assessment of each QUADAS domain can be found in the Supporting information (S2 Fig).

### Detection of SARS-CoV-2 infection

Overall, 38 data sets were excluded from the meta-analysis, as they included fewer than 20 RT-PCR positive samples. An additional 28 data sets were missing either sensitivity or specificity and were only considered for univariate analyses. The remaining 267 data sets, evaluating 198,584 tests, provided sufficient data for bivariate analysis. The results are presented in Fig 3A–3E. Detailed results for the subgroup analysis are available in the Supporting information (S3–S7 Figs).

Including any test and type of sample, the pooled estimates of sensitivity and specificity were 72.0% (95% CI 69.8 to 74.2) and 98.9% (95% CI 98.6 to 99.1), respectively. When comparing IFU and non-IFU-conform testing, sensitivity markedly differed with 76.3% (95% CI 73.7

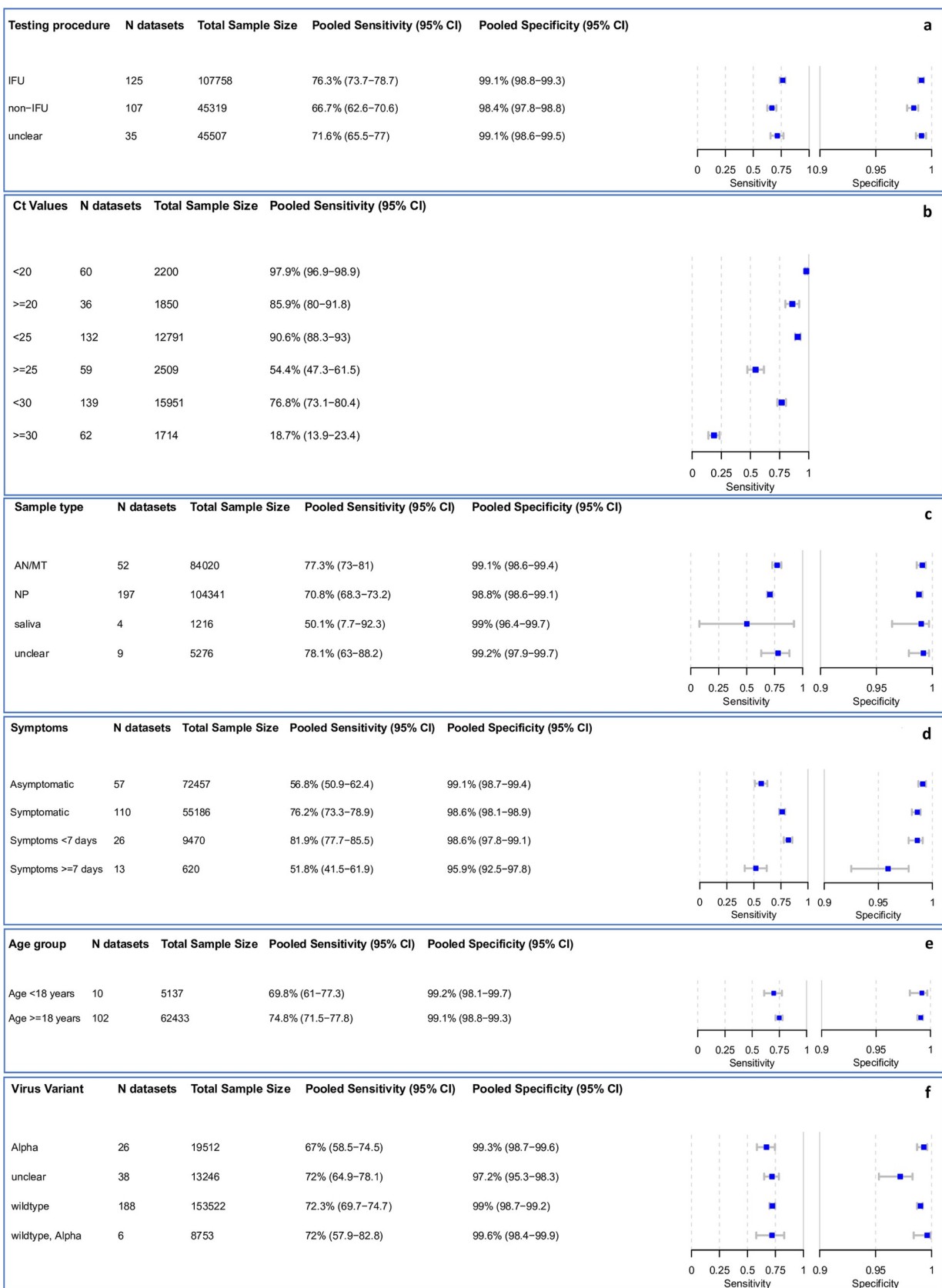

| Testing procedure | N datasets | Total Sample Size | Pooled Sensitivity (95% CI) | Pooled Specificity (95% CI) | a |
|---|---|---|---|---|---|
| IFU | 125 | 107758 | 76.3% (73.7–78.7) | 99.1% (98.8–99.3) | |
| non-IFU | 107 | 45319 | 66.7% (62.6–70.6) | 98.4% (97.8–98.8) | |
| unclear | 35 | 45507 | 71.6% (65.5–77) | 99.1% (98.6–99.5) | |

| Ct Values | N datasets | Total Sample Size | Pooled Sensitivity (95% CI) | b |
|---|---|---|---|---|
| <20 | 60 | 2200 | 97.9% (96.9–98.9) | |
| >=20 | 36 | 1850 | 85.9% (80–91.8) | |
| <25 | 132 | 12791 | 90.6% (88.3–93) | |
| >=25 | 59 | 2509 | 54.4% (47.3–61.5) | |
| <30 | 139 | 15951 | 76.8% (73.1–80.4) | |
| >=30 | 62 | 1714 | 18.7% (13.9–23.4) | |

| Sample type | N datasets | Total Sample Size | Pooled Sensitivity (95% CI) | Pooled Specificity (95% CI) | c |
|---|---|---|---|---|---|
| AN/MT | 52 | 84020 | 77.3% (73–81) | 99.1% (98.6–99.4) | |
| NP | 197 | 104341 | 70.8% (68.3–73.2) | 98.8% (98.6–99.1) | |
| saliva | 4 | 1216 | 50.1% (7.7–92.3) | 99% (96.4–99.7) | |
| unclear | 9 | 5276 | 78.1% (63–88.2) | 99.2% (97.9–99.7) | |

| Symptoms | N datasets | Total Sample Size | Pooled Sensitivity (95% CI) | Pooled Specificity (95% CI) | d |
|---|---|---|---|---|---|
| Asymptomatic | 57 | 72457 | 56.8% (50.9–62.4) | 99.1% (98.7–99.4) | |
| Symptomatic | 110 | 55186 | 76.2% (73.3–78.9) | 98.6% (98.1–98.9) | |
| Symptoms <7 days | 26 | 9470 | 81.9% (77.7–85.5) | 98.6% (97.8–99.1) | |
| Symptoms >=7 days | 13 | 620 | 51.8% (41.5–61.9) | 95.9% (92.5–97.8) | |

| Age group | N datasets | Total Sample Size | Pooled Sensitivity (95% CI) | Pooled Specificity (95% CI) | e |
|---|---|---|---|---|---|
| Age <18 years | 10 | 5137 | 69.8% (61–77.3) | 99.2% (98.1–99.7) | |
| Age >=18 years | 102 | 62433 | 74.8% (71.5–77.8) | 99.1% (98.8–99.3) | |

| Virus Variant | N datasets | Total Sample Size | Pooled Sensitivity (95% CI) | Pooled Specificity (95% CI) | f |
|---|---|---|---|---|---|
| Alpha | 26 | 19512 | 67% (58.5–74.5) | 99.3% (98.7–99.6) | |
| unclear | 38 | 13246 | 72% (64.9–78.1) | 97.2% (95.3–98.3) | |
| wildtype | 188 | 153522 | 72.3% (69.7–74.7) | 99% (98.7–99.2) | |
| wildtype, Alpha | 6 | 8753 | 72% (57.9–82.8) | 99.6% (98.4–99.9) | |

**Fig 3. (a–f) Pooled sensitivity and specificity by IFU conformity, Ct-value*, sample type, symptom status, duration of symptoms, and age.**
*Low Ct-values are the RT-PCR semiquantitative correlate for a high virus concentration, only sensitivity calculated. AN, anterior nasal; CI, confidence interval; IFU, instructions for use; MT, mid-turbinate; N, number of; NP, nasopharyngeal; RT-PCR, reverse transcription polymerase chain reaction.

to 78.7) compared to 66.7% (95% CI 62.6 to 70.6), respectively. Pooled specificity was similar in both groups: 99.1% (95% CI 98.8 to 99.3) and 98.4% (95% CI 97.8 to 98.8), respectively (Fig 3A).

## Subgroup analysis by Ct-value

We use Ct-value as a semiquantitative correlate for the sample's viral load [12]. As a point of reference, we assume as a median conversion that a Ct-value of 25 corresponds to a viral load of $1.5 * 10^6$ RNA copies per milliliter of transport media, but this varies between the types of RT-PCRs used for measuring viral load [144,215].

In samples with Ct-values <20, a very high estimate of sensitivity was found (97.9% [95% CI 96.9 to 98.9]). The pooled sensitivity for Ct-values <25 was markedly better at 90.6% (95% CI 88.3 to 93.0) compared to the group with Ct ≥ 25 at 54.4% (95% CI 47.3 to 61.5). A similar pattern was observed when the Ct-values were analyzed using cutoffs <30 or ≥30, resulting in an estimated sensitivity of 76.8% (95% CI 73.1 to 80.4) and 18.7% (95% CI 13.9 to 23.4), respectively (Fig 3B).

When pooling Ct-value estimates for TP Ag-RDT results (TP; 5,083 samples, 69 data sets) and FN (2,390 samples, 76 data sets) Ag-RDT results, the mean Ct-values were 22.2 (95% CI 21.5 to 22.8) and 30.2 (95% CI 29.6 to 30.9), respectively (S8 Fig). Across both TP and FN samples, mean Ct-value was 26.3 (95% CI 25.5 to 27.1). This demonstrates that RT-PCR positive samples missed by Ag-RDT have a substantially lower viral load (higher Ct-value) compared to those that were detected. Individual forest plots for each data set with mean Ct-values are presented in the Supporting information (S9 Fig).

## Subgroup analysis by sample type

Most data sets evaluated NP or combined NP/OP swabs (197 data sets and 104,341 samples) as the sample type for the Ag-RDT. NP or combined NP/OP swabs achieved a pooled sensitivity of 70.8% (95% CI 68.3 to 73.2) and specificity of 98.8% (95% CI 98.6 to 99.1). Data sets that used AN/MT swabs for Ag-RDTs (52 data sets and 84,020 samples) showed a summary estimate for sensitivity of 77.3% (95% CI 73.0 to 81.0) and specificity of 99.1% (95% CI 98.6 to 99.4). However, 2 studies that reported direct head-to-head comparison of NP and AN/MT samples from the same participants using the same Ag-RDT (Standard Q) reported equivalent performance [116,117]. In contrast, saliva swabs (4 data sets, 1,216 samples) showed the lowest pooled sensitivity with only 50.1% (95% CI 7.7 to 92.3) (Fig 3C). In 3 of the data sets utilizing a saliva sample, saliva was collected as whole mouth fluid (sensitivity from 8.1% [95% CI 2.7 to 17.8] to 55.6% [95% CI 35.3 to 74.5]) [24,92,154]. The fourth used a cheek swab for sample collection (sensitivity 100% [95% CI 90.3 to 100]) [55].

Due to only 3 data sets with 3,586 samples, we were not able to estimate pooled sensitivity and specificity for OP samples. Median sensitivity and specificity were 59.4% (range 50.0% to 81.0%) and 99.1% (range 99.0% to 100.0%), respectively. We were also not able to perform a subgroup meta-analysis for BAL/TW due to insufficient data, with only 1 study with 73 samples evaluating the Biocredit Covid-19 Antigen rapid test kit (RapiGEN, South Korea; henceforth called Rapigen), Panbio and Standard Q available and sensitivity ranging between 33.3%

and 88.1% [155]. However, the use of BAL/TW sampling would be considered not IFU-conforming.

### Subgroup analysis in symptomatic and asymptomatic participants

Within the data sets possible to meta-analyze, 55,186 (43.2%) samples were from symptomatic and 72,457 (56.8%) from asymptomatic persons. The pooled sensitivity for symptomatic persons was markedly higher compared to asymptomatic persons with 76.2% (95% CI 73.3 to 78.9) versus 56.8% (95% CI 50.9 to 62.4). Specificity was above 98.6% for both groups (Fig 3D).

### Subgroup analysis comparing symptom duration

Data were analyzed for 9,470 persons from 26 data sets with symptoms less than 7 days, while for persons with symptoms ≥7 days, fewer data were available (620 persons, 13 data sets). The pooled sensitivity estimate for individuals with symptoms <7 days was 81.9% (95% CI 77.7 to 85.5), which is markedly higher than the 51.8% (95% CI 41.5 to 61.9) sensitivity for individuals tested ≥7 days from onset of symptoms (Fig 3D).

### Subgroup analysis by virus variant

The 188 data sets with 153,522 samples were conducted in settings where the SARS-CoV-2 wild type was dominant. Here, sensitivity was 72.3% (95% CI 69.7 to 74.7) and specificity was 99.0% (95% CI 98.7 to 99.2). When the alpha variant (26 data sets, 19,512 samples) was the main variant, sensitivity slightly decreased to 67.0% (95% CI 58.5 to 74.5), but with overlapping CIs, and specificity remained similar (99.3% [95% CI 98.7 to 99.6]). In settings where the wild type and the alpha variant were codominant (6 data sets, 8,753 samples), sensitivity and specificity were 72.0% (95% CI 57.9 to 82.8) and 99.6% (95% CI 98.4 to 99.9), respectively.

Data were also available for the Beta, Gamma, Delta, Epsilon, Eta, and Kappa variant, but too limited to meta-analyze. Of these, most data were available for the Gamma variant, with sensitivity ranging from 84.6% to 89.9% (3 data sets, 886 samples) [202,209,213]. The main virus variant for each data set is listed in the Supporting information (S2 Table). All studies included in this review were conducted before the occurrence of the Omicron variant.

### Subgroup analysis by age

For adults (age group ≥18), it was possible to pool estimates across 62,433 samples, whereas the pediatric group (age group <18) included 5,137 samples. There was only a small difference with overlapping CIs in sensitivity with 74.8% (95% CI 71.5 to 77.8) and 69.8% (95% CI 61.0 to 77.3) for the adult and pediatric group, respectively. For those data sets that reported a median Ct-value per age group, the Ct-value was slightly lower in the adult (median 22.6, Q1 = 20.5, Q3 = 24.6, 48 data sets) compared to the pediatric group (median 23.2, Q1 = 20.3, Q3 = 25.2, 3 data sets). Specificity was similar in both groups with over 99% (Fig 3E).

### Meta-regression

The first analysis, assessing all variables that could influence sensitivity (symptom status, testing procedure [IFU-conforming versus not IFU-conforming], and mean Ct-value), included 65 data sets of symptomatic and 18 of asymptomatic persons. The second and third analysis assessed only symptomatic persons with 28 and 50 data sets, respectively. The full list of data sets for each analysis and detailed results are available in the Supporting information (Tables A–D in S4 Text).

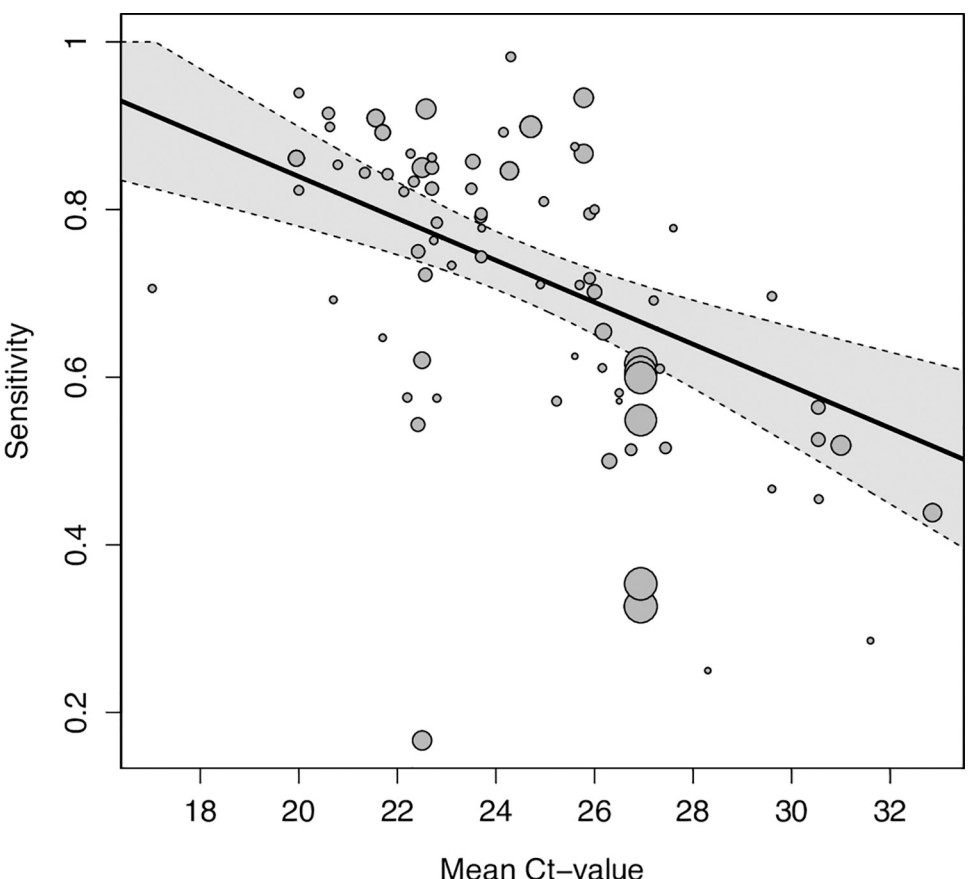

**Fig 4. Pooled estimate of sensitivity across mean Ct-values holding symptom status and IFU-status constant at their respective means.** Dotted lines are the corresponding 95% CIs. The size of each point is a function of the weight of the data set in the model, where larger data sets have larger points. CI, confidence interval; Ct, cycle threshold; IFU, instructions for use.

In the first analysis, we found viral load (as estimated by Ct-value) to be the driving factor of sensitivity. Sensitivity was estimated to increase by 2.9 percentage points (95% CI 1.7 to 4.0) for every unit the mean Ct-value decreased (Table B in S4 Text), after adjusting for symptom status and testing procedure (Fig 4). In addition, sensitivity was estimated to be 20.0 percentage points (95% CI 13.7 to 26.3) higher for samples from symptomatic compared to asymptomatic participants. However, when controlling for testing procedure and mean Ct-value, this difference declined to only 11.1 percentage points (95% CI 4.8 to 17.4). The difference between IFU-conforming versus not IFU-conforming testing procedure was not significant (5.2 percentage points [95% CI –2.6 to 13.0] higher for IFU-conforming) after controlling for symptom status and mean Ct-value.

When assessing only symptomatic participants, test sensitivity was estimated to decrease by 3.2 percentage points (95% CI –1.5 to 7.9) for every 1 day increase in average duration of symptoms (mean duration of symptoms ranged from 2.75 to 6.47 days). However, with the CI including the value 0, this effect was not statistically significant. When controlling for mean Ct-value and testing procedure, the estimated effect of the average duration of symptoms was close to 0 (0.7 percentage points [95% CI –5.0 to 6.4], Table C in S4 Text).

Concordantly, for samples collected after 7 days of symptom onset sensitivity were estimated to be 22.9 percentage points (95% CI 10.3 to 35.4) lower compared to those collected

within 7 days. When controlling for mean Ct-value and testing procedure, the model still showed a decrease in sensitivity for samples collected after 7 days of symptom onset, but again closer to 0 and no longer statistically significant (–13.8 percentage points [95% CI –27.7 to 0.1], Table D in S4 Text).

## Analysis of individual tests

Based on 179 data sets with 143,803 tests performed, we were able to perform bivariate meta-analysis of the sensitivity and specificity for 12 different Ag-RDTs (Fig 5). Across these, pooled estimates of sensitivity and specificity on all samples were 71.6% (95% CI 69.0 to 74.1) and 99.0% (95% CI 98.8 to 99.2), which were very similar to the overall pooled estimate across all meta-analyzed data sets (72.0% and 98.9%, above).

The highest pooled sensitivity was found for the SARS-CoV-2 Antigen Test (LumiraDx, UK; henceforth called LumiraDx) and the Standard Q nasal test (SD Biosensor, South Korea; distributed in Europe by Roche, Germany; henceforth called Standard Q nasal) with 82.7% (95% CI 73.2 to 89.4) and 81.4% (95% CI 73.8 to 87.2), respectively. However, all tests except the COVID-19 Ag Respi-Strip (Coris BioConcept, Belgium; henceforth called Coris; sensitivity 48.4% [95% CI 36.1 to 61.0]) had CIs that were overlapping. The pooled specificity was above 98% for all of the tests, except for the Standard F test (SD Biosensor, South Korea; henceforth called Standard F) and LumiraDx with specificities of 97.9% (95% CI 96.9 to 98.5) and 96.9% (95% CI 94.4 to 98.3), respectively. Hierarchical summary receiver operating characteristic for LumiraDx and Standard Q nasal are available in the Supporting information (S10 Fig).

For 2 Ag-RDTs, we were only able to perform a univariate analysis, due to insufficient data. Sensitivities for the COVID-19 Rapid Antigen Test Cassette (SureScreen, UK; henceforth

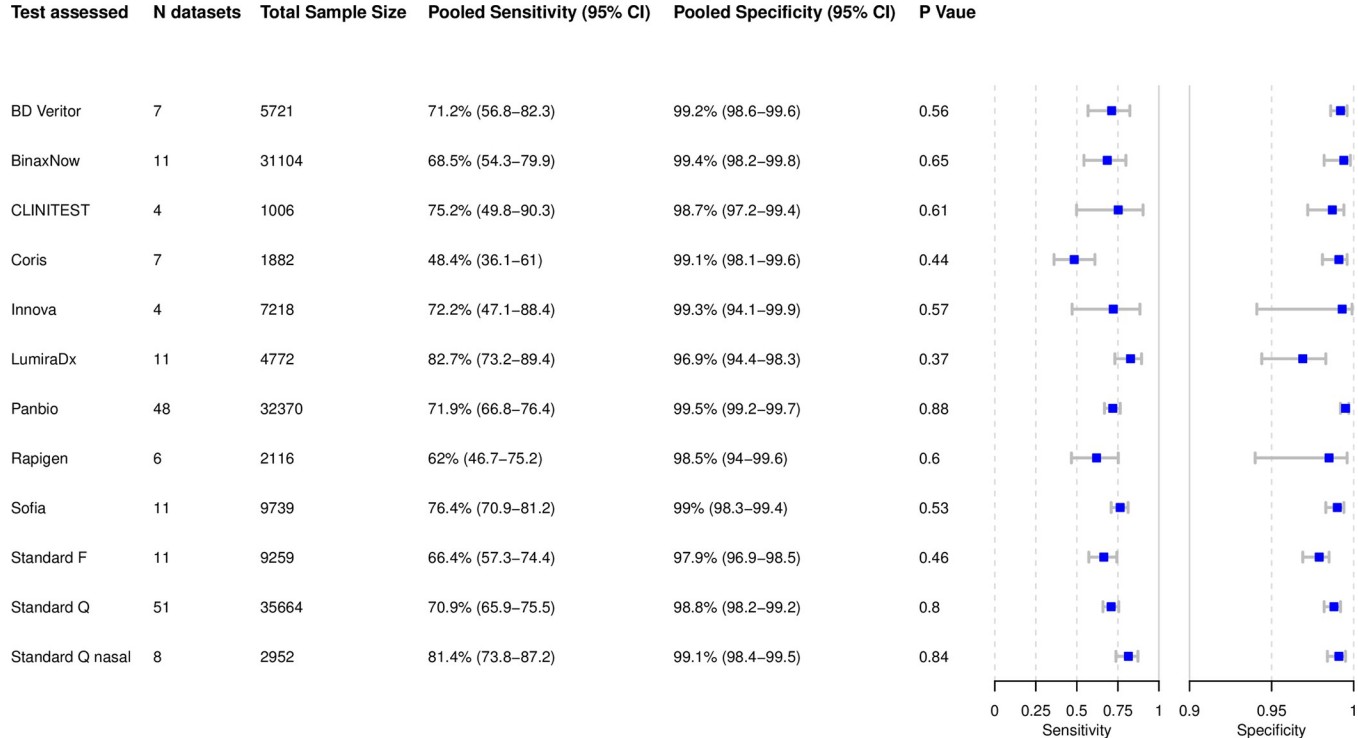

| Test assessed | N datasets | Total Sample Size | Pooled Sensitivity (95% CI) | Pooled Specificity (95% CI) | P Vaue |
|---|---|---|---|---|---|
| BD Veritor | 7 | 5721 | 71.2% (56.8–82.3) | 99.2% (98.6–99.6) | 0.56 |
| BinaxNow | 11 | 31104 | 68.5% (54.3–79.9) | 99.4% (98.2–99.8) | 0.65 |
| CLINITEST | 4 | 1006 | 75.2% (49.8–90.3) | 98.7% (97.2–99.4) | 0.61 |
| Coris | 7 | 1882 | 48.4% (36.1–61) | 99.1% (98.1–99.6) | 0.44 |
| Innova | 4 | 7218 | 72.2% (47.1–88.4) | 99.3% (94.1–99.9) | 0.57 |
| LumiraDx | 11 | 4772 | 82.7% (73.2–89.4) | 96.9% (94.4–98.3) | 0.37 |
| Panbio | 48 | 32370 | 71.9% (66.8–76.4) | 99.5% (99.2–99.7) | 0.88 |
| Rapigen | 6 | 2116 | 62% (46.7–75.2) | 98.5% (94–99.6) | 0.6 |
| Sofia | 11 | 9739 | 76.4% (70.9–81.2) | 99% (98.3–99.4) | 0.53 |
| Standard F | 11 | 9259 | 66.4% (57.3–74.4) | 97.9% (96.9–98.5) | 0.46 |
| Standard Q | 51 | 35664 | 70.9% (65.9–75.5) | 98.8% (98.2–99.2) | 0.8 |
| Standard Q nasal | 8 | 2952 | 81.4% (73.8–87.2) | 99.1% (98.4–99.5) | 0.84 |

**Fig 5. Bivariate analysis of 12 Ag-RDTs.** Pooled sensitivity and specificity were calculated based on reported sample size, true positives, true negatives, false positives, and false negatives. Ag-RDT, antigen rapid diagnostic test; CI, confidence interval; N, number of.

called SureScreen V) and the Nadal COVID-19 Ag Test (Nal von Minden, Germany; henceforth called Nadal) were similar with 57.7% (95% CI 40.9 to 74.4) and 56.6% (95% CI 26.9 to 86.3), respectively (S11 Fig). Specificity was only possible to calculate for the Nadal, which was lowest throughout the per test analysis with 91.1% (95% CI 80.2 to 100). For the remaining 62 Ag-RDTs, there were insufficient numbers of data sets for a uni-or bivariate meta-analysis. However, performance estimates and factors potentially influencing these are descriptively analyzed in the Supporting information (S4 Table) for each of the 62 tests.

For Panbio and Standard Q, it was also possible to pool sensitivity per Ct-value subgroup for each individual test. Panbio and Standard Q reached sensitivities of 97.2% (95% CI 95.3 to 99.2) and 98.1% (95% CI 96.3 to 99.9) for Ct-value <20, 89.8% (95% CI 85.4 to 94.3) and 92.6% (95% CI 88.5 to 96.7) for Ct-value <25 and 73.7% (95% CI 66.0 to 81.3) and 75.7% (95% CI 67.9 to 83.4) for Ct-value <30, respectively. For Ct-value ≥20 sensitivities for Panbio and Standard Q were 89.2% (95% CI 82.1 to 96.3) and 89.0% (95% CI 81.0 to 96.9), 51.2% (95% CI 39.4 to 63.0) and 56.4% (95% CI 45.1 to 67.8) for Ct-value ≥25, and 22.8% (95% CI 12.2 to 33.4) and 20.4% (95% CI 10.5 to 30.3) for Ct-value ≥30, respectively (S4A–S4F Fig). For Binax-Now (Abbott Rapid Diagnostics, Germany), LumiraDx, SD Biosensor, Standard F, Coris, and INNOVA SARS-CoV-2 Antigen Rapid Qualitative Test (Innova Medical Group, United States of America; henceforth called Innova), sufficient data to pool sensitivity was only available for certain Ct-values, which are available in the Supporting information (S4A–S4F Fig) as well. In addition, for 8 tests it was possible to calculate pooled sensitivity and specificity estimates only including data sets that conformed to the IFU. These are also listed in the Supporting information (S5 Table).

In total, 31 studies accounting for 106 data sets conducted head-to-head clinical accuracy evaluations of different tests using the same sample(s) from the same participant. These data sets are outlined in the Supporting information (S2 Table). Nine studies performed their head-to-head evaluation as per IFU and on symptomatic individuals. Across 4 studies, the Standard Q nasal (sensitivity 80.5% to 91.2%) and the Standard Q (sensitivity 73.2% to 91.2%) showed a similar range of sensitivity [116,130,216]. One study reported a sensitivity of 60.4% (95% CI 54.9 to 65.6) for the Standard Q and 56.8% (95% CI 51.3 to 62.2) for the Panbio in a mixed study population of symptomatic, asymptomatic, and high-risk contact persons [190]. Another study described a sensitivity of 56.4% (95% CI 44.7 to 67.6) for the Rapigen and 52.6% (95% CI 40.9 to 64) for the SGTi-flex COVID-19 Ag (Sugentech, South Korea) [164]. One study included only very few samples and using a not IFU-conforming sample type (BAL), limiting the ability to draw conclusions from the results [155].

## Publication bias

The results of the Deeks' test for all data sets with complete results ($p = 0.24$), Standard Q publications ($p = 0.39$), Panbio publications ($p = 0.81$), and Lumira ($p = 0.61$) demonstrate no significant asymmetry in the funnel plots, which suggests no publication bias. All funnel plots are listed in the Supporting information (S12 Fig)

## Sensitivity analysis

We performed 3 sensitivity analyses including 213 data sets for non-case–control studies, 216 data sets including only peer-reviewed studies, and 190 data sets including only data sets without any manufacturer influence. When excluding case–control studies, the sensitivity and specificity remained at 71.9% (95% CI 69.4 to 74.2) and 99.0% (95% CI 98.8 to 99.2), respectively. Similarly, when assessing only peer-reviewed studies, sensitivity and specificity did not change significantly with 71.1% (95% CI 68.5 to 73.6) and 98.9% (95% CI 98.6 to 99.1),

respectively. If studies that could have potentially been influenced by test manufacturers were excluded, sensitivity decreased marginally, but with overlapping CIs (sensitivity of 70.3% [95% CI 67.6 to 72.9] and specificity of 99.0% [95% CI 98.7 to 99.2]).

## Discussion

After reviewing 194 clinical accuracy studies, we found Ag-RDTs to be 76.3% (95% CI 73.7 to 78.7) sensitive and 99.1% (95% CI 98.8 to 99.3) specific in detecting SARS-CoV-2 compared to RT-PCR when performed according to manufacturers' instructions. While sensitivity was higher in symptomatic compared to asymptomatic persons, especially when persons were still within the first week of symptom onset, the main driver behind test sensitivity was a sample's viral load. LumiraDx and Standard Q nasal were the most accurate tests, but heterogeneity in the design of the studies evaluating these tests potentially favored test specific estimates.

Using Ct-value as a semiquantitative correlate for viral load, there was a significant correlation between test sensitivity and viral load, with sensitivity increasing by 2.9 percentage points for every unit decrease in mean Ct-value when controlling for symptom status and testing procedure. The pooled Ct-value for TP was on average over 8 points lower than for FN results (Ct-value of 22.2 for TP compared to 30.2 for FN results). Viral load being the deciding factor for test sensitivity confirms prior work [12].

Furthermore, sensitivity was found to be higher when samples were from symptomatic (76.2% sensitivity) compared to asymptomatic participants (56.8% sensitivity). This was confirmed in the regression model, estimating sensitivity to be 20.0 percentage points higher in samples that originated from symptomatic participants. In our previous analysis, we assumed that the increase in sensitivity is not due to the symptom status as such, but results from the fact that in symptomatic study, populations chances are higher to include participants at the beginning of the disease with high viral load [4]. In the present analysis, this assumption shows to be largely true. Controlling for Ct-value, the RT-PCR correlate for viral load, the effect of symptomatic versus asymptomatic participants on test sensitivity strongly decreased to 11.1 percentage points. As others found symptomatic and asymptomatic individuals to have the same viral load when at the same stage of the disease [8], we would have expected the regression coefficient to have decreased even further to 0. This nonzero difference in sensitivity between symptomatic and asymptomatic participants may be due to the lack of access to individual participant Ct-values, which required our analyses to control for the mean Ct-value over all participants in a data set rather than the individual Ct-values. Furthermore, some variability is likely introduced by the testing for the Ag-RDT and the RT-PCR not to occur from the sample. Therefore, some degree of residual confounding is likely present.

We also found sensitivity to be higher when participants were tested within 7 days of symptom onset (81.9% sensitivity) compared to >7 days (51.8% sensitivity). Concordantly, our regression model estimated that sensitivity decreases by 3.2 percentage points for every 1-day increase in mean symptom duration. Again, this decrease in sensitivity is driven by viral load as was seen when controlling for Ct-value. Importantly, it is not yet clear how the emergence of new SARS-CoV-2 VoC and the growing vaccination coverage will affect Ag-RDTs sensitivity in the early days after symptom onset. Most of the studies included in this analysis were performed at the time the wild type and Alpha variant were circulating. Test sensitivity was slightly lower for the Alpha variant compared to the wild type (67.0% [95% CI 58.5 to 74.5] versus 72.3% [95% CI 69.7 to 74.7]). However, conclusions on differences in performance between variants are difficult to draw as between study heterogeneity was substantial and, while this does not preclude a difference between groups, CIs were widely overlapping. Furthermore, pooled sensitivity for studies where Alpha and wild type were codominant (72.0%

[95% CI 57.9 to 82.8]) were similar to that of the wild type alone. Similar Ag-RDT sensitivity was also found with the Delta variant compared to wild type, and for the Omicron SARS-CoV-2 variant initial data suggests similar clinical performance as well, although analytical performance pointed toward a potentially lower performance [217–220]. Vaccination did not affect viral kinetics in the first week [221] and is unlikely to do so for the Omicron variant [222]. To further inform public health decision makers on the best strategy to apply Ag-RDTs, clinical accuracy studies in settings with high prevalence of the Omicron variant are urgently needed.

Looking at specific tests, LumiraDx and Standard Q nasal showed the highest sensitivity, performing above the 80% sensitivity target defined by WHO. However, while the Standard Q nasal was 99.1% (95% CI 98.4 to 99.5) specific, the LumiraDx only had a specificity of 96.9% (95% CI 94.4 to 98.3), which is just below the WHO target of 97%. The reason for the lower specificity is unclear, particularly as independent analytical studies also confirmed the test had no cross-reactivity [106]. Sample to sample variability must be considered, particularly as the sensitivity of the index tests approaches that of the reference test. The 2 most often evaluated tests, namely Panbio (32,370 samples, sensitivity of 71.9%) and Standard Q (35,664 samples, sensitivity of 70.9%), performed slightly below the overall average. Similarly, Panbio and Standard Q were also the most extensively evaluated Ag-RDTs in the prior analysis, and with a sensitivity slightly above average [4]. Nonetheless, this updated analysis indicates that limited added value is to be expected from any further analysis of Ag-RDTs' overall sensitivity or the sensitivity of the most widely evaluated tests. However, it will be important to continue to reassess tests' analytical sensitivity for detection of new specific variants (e.g., Omicron). In addition, with a recent WHO guideline on self-performed Ag-RDTs having laid the scientific foundation [223], it would be of interest to further evaluate the accuracy and ease of use of self-performed Ag-RDTs, or specific characteristics of instrument-based Ag-RDTs.

Furthermore, sensitivity strongly differed between studies that conducted the Ag-RDTs as per manufacturer's instructions and those that did not (sensitivity of 66.7% for not IFU-conforming versus 76.3% for IFU-conforming). This was also reflected in our regression model, where test performance decreased when not following manufacturer's instructions; however, this was not significant (−5.2 percentage points [95% CI −13.0 to 2.6]). In regards to sample types, saliva showed a markedly lower sensitivity of 50.1%, compared to NP or AN/MT samples, confirming what we found in our previous analysis [4]. Especially in light of the current debate on whether saliva or throat swabs might be a more sensitive sample to detect the SARS-CoV-2 Omicron variant than NP or AN/MT samples [224–226], further research is urgently needed to quantify the difference in viral load resulting from different sample types and thus the effect of sample type on test sensitivity.

In concordance with the above, many studies reporting an unusually low sensitivity performed the Ag-RDT not as per IFU [30,32,44,70,101,112,137,188] or used saliva samples [24,154,159,227]. However, 2 studies with IFU-conforming testing procedure on NP or AN/MT sample still showed a low sensitivity. This quite likely results from the on average low viral load in 1 study [53] and the asymptomatic study population in the other [179]. On the contrary, compared to the other studies unusual high sensitivity was found in studies where average viral load was high [49,88,148,149] or participants were mainly within the first week of symptom onset [46,58,139].

The main strength of our study lies in its living approach. The ability to update our methodology as the body of evidence grows has enabled an improved analysis. For example, while data were too heterogenous for a meta-regression during the prior analysis, with additional data sets we are now able to analyze the relationship between an Ag-RDT's sensitivity, the samples' Ct-value, and the participants' symptom status in depth. Similarly, we decided to focus on

clinical accuracy studies for POC Ag-RDTs in this current review as analytical accuracy studies require a dedicated approach to be comparable. Furthermore, the main results of our latest extractions are publicly available on our website. This has not only equipped public health professionals with an up-to-date overview on the current research landscape [228,229], but also led other researchers and the test manufacturers to check our data, improving the quality of our report through continuous peer-review.

Nonetheless, our study is limited in that we use RT-PCR as a reference standard to assess the accuracy of Ag-RDTs, which are generally much more sensitive than Ag-RDTs [230] and might be a less appropriate reference standard than viral culture [139,231,232]. However, viral culture is available in research settings only and its validity as a true proxy of actual transmissibility is not proven; therefore, we find RT-PCR a suitable reference standard for the clinical accuracy studies included in this review. Furthermore, we fully acknowledge that Ct-value is only an estimate of viral load, and that the correlation between Ct-value and viral load varies between RT-PCR assays, potentially affecting the sensitivity and specificity of the evaluated Ag-RDTs [215]; nonetheless, we believe that the analysis of pooled Ct-value data across a very large data set is a useful strategy to understand the overall contribution of viral load to Ag-RDT performance. Moreover, we are aware that the test specific sensitivities and specificities can be influenced by differences in study design. However, we aimed to counterbalance this effect by assessing relevant aspects in study design for each study and analyzing outliers. To enhance comparability in between clinical accuracy studies, future studies should include individuals at a similar stage in the disease, use the same sample types, and adhere to the WHO standard for measuring SARS-CoV-2 viral load [13]. Finally, our study only includes literature up until August 31, 2021. Thus, we were not able to analyze information on Delta or Omicron variants, and look to future research to close this gap in literature.

## Conclusions

In summary, Ag-RDTs detect most of the persons infected with SARS-CoV-2 when performed according to the manufacturers' instructions. While this confirms the results of our previous analysis, the present analysis highlights that the sample's viral load is the most influential factor underlying test sensitivity. Thus, Ag-RDTs can play a vital role in detecting persons with high viral load and therefore likely to be at highest risk of transmitting the virus. This holds true even in the absence of patient symptoms or differences in the duration of symptoms. To foster further research analyzing specific Ag-RDTs and the factors influencing their sensitivity in more detail, standardization of clinical accuracy studies and access to patient level Ct-value and duration of symptoms are essential.

## Supporting information

**S1 PRISMA Checklist. PRISMA checklist.**
(DOCX)

**S1 Fig. Forest plots of all Ag-RDTs.** Ag-RDT, antigen rapid diagnostic test; CI, confidence interval; FN, false negative; FP, false positive; TN, true negative; TP, true positive.
(PDF)

**S2 Fig. Details of QUADAS assessment.**
(PDF)

**S3 Fig. Forest plots for subgroup analysis by Ct-values.** CI, confidence interval; Ct, cycle threshold.
(PDF)

**S4 Fig. Forest plots for subgroup analysis by Ct-values per test.** CI, confidence interval; Ct, cycle threshold.
(PDF)

**S5 Fig. Forest plots for subgroup analysis by IFU versus non-IFU.** CI, confidence interval; FN, false negative; FP, false positive; IFU, instructions for use; TN, true negative; TP, true positive.
(PDF)

**S6 Fig. Forest plots for subgroup analysis by sample type.** CI, confidence interval; FN, false negative; FP, false positive; TN, true negative; TP, true positive.
(PDF)

**S7 Fig. Forest plots for subgroup analysis by symptomatic versus asymptomatic.** CI, confidence interval.
(PDF)

**S8 Fig. Forest plot for subgroup analysis by mean Ct-values for TP and FN samples.** CI, confidence interval; Ct, cycle threshold; FN, false negative; TP, true positive.
(PDF)

**S9 Fig. Forest plots for subgroup analysis by mean Ct-values for TP and FN samples.** CI, confidence interval; Ct, cycle threshold; FN, false negative; TP, true positive.
(PDF)

**S10 Fig. HSROC curve Standard Q nasal and LumiraDx Ag-RDT.** Ag-RDT, antigen rapid diagnostic test; HSROC, Hierarchical summary receiver-operating characteristic.
(PDF)

**S11 Fig. Forest plot for univariate analysis for Nadal and SureScreen V.** CI, confidence interval.
(PDF)

**S12 Fig. Funnel plots for all, LumiraDx, Panbio, and Standard Q studies.**
(PDF)

**S1 Table. List of parameters extracted from studies.**
(XLSX)

**S2 Table. Summary of tests.**
(XLSX)

**S3 Table. Overall and sensitivity analysis.**
(XLSX)

**S4 Table. Tests analyzed descriptively not included in meta-analysis.**
(XLSX)

**S5 Table. Test specific IFU analysis.**
(XLSX)

**S1 Text. Study protocol submitted to PROSPERO.**
(DOCX)

**S2 Text. Search strategy.**
(DOCX)

**S3 Text. QUDAS-2 assessment interpretation guide.**
(DOCX)

**S4 Text. Details meta-regression.**
(DOCX)

**S5 Text. List of studies excluded.**
(DOCX)

**S6 Text. Studies potentially influenced by the test manufacturer.**
(DOCX)

## Author Contributions

**Conceptualization:** Lukas E. Brümmer, Stephan Katzenschlager, Sean McGrath, Claudia M. Denkinger.

**Data curation:** Lukas E. Brümmer, Stephan Katzenschlager, Sean McGrath, Stephani Schmitz, Mary Gaeddert, Christian Erdmann, Marc Bota, Aurélien Macé, Berra Erkosar, Claudia M. Denkinger.

**Formal analysis:** Lukas E. Brümmer, Stephan Katzenschlager, Sean McGrath, Stephani Schmitz, Mary Gaeddert, Aurélien Macé, Berra Erkosar, Claudia M. Denkinger.

**Funding acquisition:** Claudia M. Denkinger.

**Investigation:** Lukas E. Brümmer, Stephan Katzenschlager, Sean McGrath, Stephani Schmitz, Mary Gaeddert, Christian Erdmann, Marc Bota, Claudia M. Denkinger.

**Methodology:** Lukas E. Brümmer, Stephan Katzenschlager, Sean McGrath, Claudia M. Denkinger.

**Project administration:** Lukas E. Brümmer, Stephan Katzenschlager, Claudia M. Denkinger.

**Resources:** Lukas E. Brümmer, Stephan Katzenschlager, Claudia M. Denkinger.

**Software:** Sean McGrath, Mary Gaeddert, Maurizio Grilli, Aurélien Macé, Berra Erkosar.

**Supervision:** Lukas E. Brümmer, Claudia M. Denkinger.

**Validation:** Lukas E. Brümmer, Stephan Katzenschlager, Sean McGrath, Stephani Schmitz, Claudia M. Denkinger.

**Visualization:** Lukas E. Brümmer, Stephan Katzenschlager, Sean McGrath, Mary Gaeddert, Claudia M. Denkinger.

**Writing – original draft:** Lukas E. Brümmer, Stephan Katzenschlager, Sean McGrath, Claudia M. Denkinger.

**Writing – review & editing:** Lukas E. Brümmer, Stephan Katzenschlager, Sean McGrath, Stephani Schmitz, Mary Gaeddert, Christian Erdmann, Marc Bota, Maurizio Grilli, Jan Larmann, Markus A. Weigand, Nira R. Pollock, Aurélien Macé, Berra Erkosar, Sergio Carmona, Jilian A. Sacks, Stefano Ongarello, Claudia M. Denkinger.

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
