## [Editor Report · Decision Letter 0]

10 Feb 2022

Dear Dr Denkinger, 

Thank you for submitting your manuscript entitled "Accuracy of rapid point-of-care antigen-based diagnostics for SARS-CoV-2: an updated systematic review and meta-analysis with meta regression analyzing influencing factors" for consideration by PLOS Medicine.

Your manuscript has now been evaluated by the PLOS Medicine editorial staff and I am writing to let you know that we would like to send your submission out for external assessment.

However, we first need you to complete your submission by providing the metadata that is required for full assessment. To this end, please login to Editorial Manager where you will find the paper in the 'Submissions Needing Revisions' folder on your homepage. Please click 'Revise Submission' from the Action Links and complete all additional questions in the submission questionnaire.

Please re-submit your manuscript within two working days, i.e. by Feb 14 2022 11:59PM.

Once your full submission is complete, your paper will undergo a series of checks in preparation for full assessment.

Kind regards,

Richard Turner, PhD

plosmedicine@plos.org

---

## [Decision Letter · Decision Letter 1]

8 Mar 2022

Dear Dr. Denkinger,

Thank you very much for submitting your manuscript "Accuracy of rapid point-of-care antigen-based diagnostics for SARS-CoV-2: an updated systematic review and meta-analysis with meta regression analyzing influencing factors" (PMEDICINE-D-22-00454R1) for consideration at PLOS Medicine. 

Your paper was discussed with an academic editor with relevant expertise and sent to independent reviewers, including a statistical reviewer. The reviews are appended at the bottom of this email and any accompanying reviewer attachments can be seen via the link below:

[LINK]

In light of these reviews, we will not be able to accept the manuscript for publication in the journal in its current form, but we would like to invite you to submit a revised version that addresses the reviewers' and editors' comments fully. You will appreciate that we cannot make a decision about publication until we have seen the revised manuscript and your response, and we expect to seek re-review by one or more of the reviewers. 

We hope to receive your revised manuscript by Mar 29 2022 11:59PM. Please email us (plosmedicine@plos.org) if you have any questions or concerns.

Please let me know if you have any questions, and we look forward to receiving your revised manuscript. 

Sincerely,

Richard Turner, PhD

Senior editor, PLOS Medicine

rturner@plos.org

Please update the search to the most recent date feasible. 

Our academic editor commented: "the variant specific analyses one of the reviewers mentioned may be plausible with alpha, delta, and omicron estimates if up to date data are available (would be interesting to see whether there is effect modification depending on what was the predominant variant at the time of testing)".

Please add a new final sentence to the "Methods and findings" subsection of your abstract, which should begin "Study limitations include ..." or similar and should quote 2-3 of the study's main limitations. 

In the abstract and throughout the paper, please quote p values alongside 95% CI, where available. 

After the abstract, please add a new and accessible "Author summary" section in non-identical prose. You may find it helpful to consult one or two recently-published research papers in PLOS Medicine to get a sense of the preferred style. 

Please add some additional detail in the Methods section (main text) regarding the methods for extracting studies and how disagreement between researchers was resolved, for example. 

At line 162, we note that heterogeneity was estimated "visually". Can statistical analyses be reported?

Please restructure the start of the Discussion section (main text), as the first paragraph should summarize the study's main findings.

Throughout the text, please remove spaces from within reference call-outs (e.g., "... in which they are used [2,3].").

In the reference list, please convert all italic and boldface text into plain text. 

Please list 6 authors names followed by "et al.", where appropriate. 

Noting reference 181, please use the journal name abbreviation "PLoS ONE".

Noting reference 222 and others, please ensure that all citations have full access details. 

Comments from the reviewers:

*** Reviewer #1: 

The authors describe an update/publication of their living systematic review of antigen tests, with particular emphasis on summarizing/clarifying results on subsets of patient populations. This work is important, comprehensive, and well written, and the technical details of the meta-analysis are appropriate (I caution that while I have a fairly strong grasp of the mathematics and techniques described, I would not consider myself a meta-analysis expert). The sensitivity analysis to exclude papers that might have been un/intentionally "bought" by manufacturers is a nice touch. In my opinion it is thus worthy of publication. I recommend acceptance with only minor revisions (below).

Major points

--------------

The only substantive comment regards viral load. As the authors state, and as is well known, viral load ranges widely over the course of infection and among infected individuals. Unfortunately, and to some consternation, even after two-plus years, viral loads are still not reported clinically, as they are for numerous other infections (for example, HIV, HCV), for various reasons, or in studies, where the quantitative measure used is the Ct value. This is problematic, as Ct value ranges can differ from one reference test to another, sometimes markedly.

To their credit, the authors of this study are well aware of this issue, discuss it ably, and and use viral loads in the cases where they are available (I feel their discussion in 554-555 is perhaps a bit more defensive-sounding than it needs be: Ct value is essentially the only available quantitation at present, so however imperfect, it is completely reasonable to do; I would be surprised if there were pushback to that). 

However:

--it would be of great utility to name the specific RT-qPCR assay/platform used in each assay, where available, in a supplemental document. (Apologies if this was done and I missed it; I looked ca. L283-286.) I suspect this would explain some of the outliers in Fig. 4. Ideally, the authors would add platform as a regression variable, but I respect that this might be too much to ask, so would not require

redoing the regression with platform as a condition of publication. The effect of differences in limits of detection of these platforms on sensitivity/specificity/concordance between tests should be mentioned (illustrated well in Fig. 2 of https://pubmed.ncbi.nlm.nih.gov/34076471/ , which should be cited).

--it would be useful (va. L308-319) to give the reader even a rough sense of what viral loads these Ct value cutoffs most likely correspond to, on average, if possible. If not possible, a sentence in this section declaring that to be the case, explaining that Ct scales vary (their reference 3 is a suitable citation). My hope is the field will at some point move to viral loads, and it would wonderful to "future-proof" this excellent review, simply by adding such a Ct-to-viral-load conversion anchor (even a rough one, for example, plus-or-minus two or three orders of magnitude).

Hopefully these (and the minor comments below) will be straightforward points for the authors to address, further strengthening this exceptional study ahead of publication.

Minor comments

------------------

Given that it is known what the predominant strain was in each geography (for example https://github.com/hodcroftlab/covariants), is it not possible to list the likely strain for each study? That would offer the opportunity to regress by strain in the future.

L99-110, PRISMA checklist: I missed whether the search was done independently by two individuals. This may be a requirement of MOOSE but not PRISMA, or other journals, but not PRISMA/PLoS Medicine. If it was done, please mention. If it was not, that is also fine. (I am *not* suggesting it be done, if it wasn't, as a condition for publication.)

L228-230: Were the non-English-language papers included? I assume yes, but please clarify.

L151-155: I did not quite follow whether this "relaxing" allowed/resulted in overlap, or just aligned bin edges. Perhaps the authors could clarify.

L302-303: the 95% CIs overlap. Would the authors comment on the probability that this increase is real? I ask especially since the 95% CIs do not overlap for the specificities, yet the authors do not claim that the specificity was lower when using the IFU (which would not be unexpected, given a lower sensitivity). The comparator adjective should be used similarly: I would suggest either these are both comparable, or the first is comparable (since the 95% CIs overlap) and the second is lower. I would suggest they are both comparable.

L328-331: It is interesting that saliva swabs showed the lowest sensitivity, since saliva has been shown to be comparable to NP as a specimen type (https://pubmed.ncbi.nlm.nih.gov/34406838/). The 95%CI range for saliva in Fig. 3 raises the same question (5% to 95%!). Was there/was it possible to detect a difference between the swabbed saliva and the whole-mouth saliva samples? 

Fig. 5: Given the narrow range of specificities, might the x-axis be usefully contracted to 0.85 or 0.9-1.0, so that differences might be visualized?

L551-552: Would also cite https://www.medrxiv.org/content/10.1101/2021.12.22.21268274v1

*** Reviewer #2: 

Brümmer and colleagues present a systematic review and meta-analysis of the diagnostic accuracy of rapid antigen tests for SARS-CoV-2 compared to a gold standard RT-PCR. Across 194 studies that could be meta-analyzed, the overall sensitivity was 72% and specificity was 98.9%. The study builds upon a previous living systematic review by adding additional studies, though overall findings remain near identical. The added value of this study is in its meta-regression and stratified analyses, which help clarify the impact of various factors of Ag-RDT sensitivity/specificity. The methods used are rigorous and search thorough. I do have some suggestions for the authors to better describe the performance of Ag-RDTs:

* I believe the IFU evaluation is valuable, but think there is likely more that can be done, if the data is available. The value of Ag-RDTs can only be fully realized if they can be performed without aid or observation of health workers. The authors specifically mention on lines 523-524 that future research could evaluate "Ag-RDTs that are self-performed or those that are instrument-based." It would be interesting to know if any of the 194 studies were self-performed and/or self-read, and if further stratified analyses could be done to evaluate if sensitivity is impacted (as compared to health care professional performed and read). This would better clarify the role and impact of Ag-RDTs as a public health tool.

* In the same vein, performance of Ag-RDT may be impacted by circulating variants. Were data available to evaluate this (e.g., for wild type, alpha, delta) within the existing cohorts in the review (understanding this study was done pre-omicron)? From the discussion section, it is clear that most studies were done during wild-type or alpha dominance. Proxies could be used such as dominating variant circulating in the study setting during the study period (which I believe is defensible as study mean age and Ct were used in meta-regression), and would permit comparison.

* With respect to specific brands of Ag-RDT being the best performers, I was left wondering if studies that used these Ag-RDT also contained participants with characteristics that increased sensitivity (e.g., IFU-conforming, symptomatic patients, early in symptoms, lower Ct), which would limit our ability to conclude which test may have highest sensitivity. I think more could be done to contextualize these findings, particularly in the discussion. Forgive me if I missed this information.

* What were the specimens collected for RT-PCR for the reference standard across studies? I was unable to find this information, however I may have missed it in the dense supplement. There are differences in sensitivity between specimen types for RT-PCR, so if there were different specimens used for the reference standard across studies this should be clear and evaluated.

* I believe there may be typos with respect to number of included studies and data sets throughout the manuscript (e.g., Fig 1 states 194 studies included, line 216 says 174). Please review.

*** Reviewer #3: 

Alex McConnachie, Statistical Review

Brummer et al present the latest update from the living review of rapid antigen tests for SARS-Cov-2. The analysis now includes meta regression, looking at factors associated with diagnostic performance. This review considers the statistical element of the paper.

Generally, the statistical methods are good, with the results presented and interpreted appropriately. I have a few comments, which are fairly minor.

As a non-clinical reader, I would have appreciated an explanation of Ct values, in the sense that a low Ct value indicates a higher viral load, earlier in the paper, or even in the abstract.

The statistical methods section of the paper is quite good, but does not mention the use of summary ROC curves, when perhaps it should. Also, line 181 mentions multivariate regression of factors associated with sensitivity; "multivariable" is a more accurate term.

There is mention of the assessment of publication bias using Deeks' test for funnel plot asymmetry. However, I could not find any funnel plots in the paper or the supplement, nor any results of these tests. Why are these not reported? Also, assessment of publication bias is usually not recommended when the number of studies is small; did the authors have specific criteria in terms of the number of studies, when it came to the assessment of publication bias?

Most research studies require a justification for the sample size of the study. For meta analyses, this is generally not an issue. However, in this case, since this is a living review, the authors have been able to see the data accumulating over time, so perhaps some sort of sample size justification is warranted. Why do the analysis now? Were there pre-defined criteria that triggered this analysis?

3 sensitivity analyses are reported. What is the rationale for doing these as sensitivity analyses (i.e. repeating the analysis after excluding a group of studies) rather than as subgroup analyses (i.e. estimating the difference in sensitivity and specificity estimates obtained from studies of different types)?

In Figures 3 and 5, it is noticeable that the pooled specificity estimates are all very high. So high, in fact, that it is very difficult to see what is going on, as all the points and confidence intervals are bunched up to the right of the figure. I can understand that there is a need to keep the axes the same, for comparability, but would it help to truncate the x-axis in these figures, e.g. with a lower range of 90%? A lower limit can be used in the supplement, but for the main body of the paper, a higher value perhaps makes sense.

*** Reviewer #4: 

Estimated Authors,

I've read the present study with great interest. Following a living review of POC antigen-based test for SARS-CoV-2 (updated until August 2021), Authors were able to summarise available evidence on sensitivity and specificity of such instruments. Because of the reduced turnaround time, as well as the option to perform such tests in settings other than hospitals and medical laboratories, POC antigen-based test represent a valuable asset in in our global efforts against SARS-CoV-2 pandemic. Unfortunately, as previously stressed by several studies, the actual sensitivity of these POC tests may be quite inappropriate to fulfil the target of promptly and accurately identify incident cases of SARS-CoV-2 infections. With a sensitivity of around 70%, a large share of cases may fail to be diagnosed, particularly in cases with a low to moderate replication of the pathogen, and in early stages of infections.

Authors, through an appropriate study design and an accurate and diligent application of a proper methodology were able to provide a valuable piece of information for all professionals interested in this specific topic.

Briefly, I've neither requests or recommendations for improving this paper, whose acceptance "as it is" I strongly recommend.

***

[LINK]

---

## [Decision Letter · Decision Letter 2]

22 Apr 2022

Dear Dr. Denkinger,

Thank you very much for re-submitting your manuscript "Accuracy of rapid point-of-care antigen-based diagnostics for SARS-CoV-2: an updated systematic review and meta-analysis with meta regression analyzing influencing factors" (PMEDICINE-D-22-00454R2) for consideration at PLOS Medicine.

I have discussed the paper with our academic editor and it was also seen again by three reviewers. I am pleased to tell you that, once the remaining editorial and production issues are fully dealt with, we expect to be able to accept the paper for publication in the journal.

[LINK]

Please let me know if you have any questions, and we look forward to receiving the revised manuscript.   

Sincerely,

Richard Turner, PhD

rturner@plos.org

Requests from Editors:

Our academic editor requests that you state in the paper, as a limitation, perhaps, that the absence of literature post-August 2021 means that information on testing for the Delta and Omicron variants, for example, is lacking. 

At line 81, please make that "... proved to be" or similar. 

At line 649, we suggest "... enabled an improved analysis". 

Please add an institutional author name to references 1 & 3, and any others.

Please spell out the group author name in reference 10 and any other relevant references.

Comments from Reviewers:

*** Reviewer #1: 

Overall the authors have addressed all the points brought up in my review, further strengthening this major contribution to the literature. The variant discussion in particular was excellent. I have now just the following minor points; if the authors make these, I recommend acceptance. In the interest of time, I would not need to see a revision again before publication (I trust the authors/editors).

L385: Would suggest the following change (additions in ALL CAPS): "Can be assumed to be"  "AS A POINT OF REFERENCE, we assume AS A MEDIAN CONVERSION that Ct value of 25 corresponds to a viral load of 1.5 * 10^6 RNA copies per milliliter OF TRANSPORT MEDIA, but [rest of sentence and refs are fine]". Gives a better idea of why the assumption was made.

L222-223: please cite Covariants.org as follows, per the author's recommendation (https://covariants.org/faq#how-should-i-cite-or-acknowledge-this-work">https://covariants.org/faq#how-should-i-cite-or-acknowledge-this-work): Emma B. Hodcroft. 2021. "CoVariants: SARS-CoV-2 Mutations and Variants of Interest." https://covariants.org/. This will help the author rationalize to funding agencies the value of this work, keeping it available to the scientific community going forward.

L662 (or possibly L175-177): It is worth noting explicitly that RT-PCR tests are generally much more sensitive than antigen tests, which fits nicely with the better correlation at lower Ct values (most COVID-19-experienced readers will know this, but safer to not assume that future/all readers will). Please cite https://academic.oup.com/cid/article/73/9/e3042/6127024 as to the importance of LoD.

Table S2: Would add a column for the limit of detection of the comparator and a column for the limit of detection of the Ag-RDT (I went through all the supplements; apologies if I missed this). These are also usefully mentioned where appropriate in L360-362.

*** Reviewer #2: 

I'd like to thank the authors for thoroughly revising their manuscript based on the editor and reviewer comments. My only remaining comment is for the authors to potentially re-consider how they speak about overlapping confidence intervals - which are not a perfect method to state two measures are not significantly different (see: https://www.tandfonline.com/doi/abs/10.1198/000313001317097960 and https://cscu.cornell.edu/wp-content/uploads/73_ci.pdf). The authors can maintain their messaging even with these statements on overlapping CIs removed. Currently, there is an implication that differences in sensitivity do not exist between tests since confidence intervals overlap - though I think the authors intend to say there is heterogeneity.

*** Reviewer #3: 

Alex McConnachie, Statistical Review

I thank the authors for their consideration of my previous comments. I am happy with the responses, and have no further comments to make.

***

[LINK]

---

## [Editor Report · Decision Letter 3]

4 May 2022

Dear Dr Denkinger, 

On behalf of my colleagues and the Academic Editor, Dr Suthar, I am pleased to inform you that we have agreed to publish your manuscript "Accuracy of rapid point-of-care antigen-based diagnostics for SARS-CoV-2: an updated systematic review and meta-analysis with meta regression analyzing influencing factors" (PMEDICINE-D-22-00454R3) in PLOS Medicine.

PRESS

Sincerely, 

Richard Turner, PhD 

rturner@plos.org